# Incidence and influencing factors related to social isolation among HIV/AIDS patients: Protocol for a systematic review and meta-analysis

**Qiao Wu**, **Jiarong Tan** *, **Shu Chen** *, **Jiayi Wang, Xiaogang Liao, Lingling Jiang**

Department of Nursing, Chongqing Public Health Medical Center, Chongqing, China

* 17345880991@163.com (JT); chenshushu1122@126.com (SC)

## Abstract

### Background

People living with HIV (PLWH) are susceptible to social isolation as a result of stigma and discrimination, which not only diminishes adherence to antiretroviral therapy but also heightens the risks of hospital readmission, depression, and mortality. However, there is currently no systematic review addressing the occurrence and impact of social isolation in individuals with HIV. Therefore, this study undertook a comprehensive systematic review and meta-analysis of existing literature to examine the prevalence and influencing factors associated with social isolation among PLWH.

### Methods and analysis

PubMed, EMBASE, CINAHL, Cochrane Library, Web of Science, Google Scholar, China Science and Technology Journal Database, The China National Knowledge Infrastructure, WanFang Data and Chinese Biomedicine Literature Database will be searched from the establishment of the database to the latest search date. Literature screening, data extraction and literature quality assessment will be done independently by two researchers and results will be cross-referenced. Data analysis will be performed using stata15.1 software. Risk of publication bias will be assessed using Begg's and Egger's methods. Heterogeneity between studies will then be assessed using the $I^2$ index and its 95% CI and Q statistics. Sources of heterogeneity will be accounted for by subgroup and sensitivity analyses.

### Results

The results may reveal the prevalence of social isolation among PLWH and provide data support for understanding its etiology and prevention.

### Conclusion

By systematically reviewing the existing literature on social isolation among PLWH, this study aims to provide a comprehensive understanding of the prevalence of social isolation within this population, elucidate the detrimental effects it poses for people affected by HIV, and

relevant data from this study will be made available upon study completion.

**Funding:** The author(s) received no specific funding for this work.

**Competing interests:** The authors have declared that no competing interests exist.

effectively inform targeted interventions for high-risk groups. Furthermore, these findings offer valuable insights to support evidence-based decision-making in public health policy.

## Systematic review registration

PROSPERO registration number: CRD42024499044

## Introduction

Social isolation, also known as social isolation, refers to an individual's lack of a sense of belonging in society, insufficient contact with others and social contacts, and a fractured or isolated state of interpersonal interactions, which leads to negative physical and/or psychological consequences [1–3]. It is an objective and quantifiable reflection of reduced social network size and paucity of social contact [4].The prevalence of social isolation is increasing in contemporary society [5]. Studies have demonstrated that more than one in ten individuals in the adult population reported social isolation [6] and nearly one-quarter of adults aged 65 years and older are considered socially isolated [7]. While social isolation may be more common in older age groups, it similarly affect younger age groups [8]. Relevant studies have shown that the incidence of social isolation in children with hearing loss is 64.7% [9].

Whereas PLWH may be more vulnerable to social isolation because they experience or perceive stigma and discrimination, which prevents them from building social network relationships [10–12]. Furthermore, HIV infection itself intensifies many normal aging processes and increases the incidence and severity of frailty, which may hinder their participation in social activities and exacerbate social isolation [13, 14]. According to the literature, the prevalence of social isolation in HIV patients can be as high as 48.5% to 59% [15, 16].

Social isolation has serious adverse effects on PLWH, which not only reducing their adherence to treatment, but also increasing their risk of hospitalization, depression and so on [17, 18]. It may even synergise with other immune processes, leading to an increased risk of morbidity and mortality in PLWH [19]. It is clear that social isolation is an important factor affecting the prognosis and quality of life of PLWH, and has become a global public health problem [20].

In addition, few viable interventions have been identified for the social isolation of PLWH [21] so early identification of risk factors associated with social isolation among PLWH, so as to take targeted interventions is an important way to prevent the occurrence of social isolation among PLWH. However, there is less information about social isolation among HIV [17], and there is no systematic retrospective study on the occurrence of social isolation and its associated factors in PLWH. Based on this, the present study was conducted to systematically evaluate the epidemiological status of social isolation in HIV patients and related influencing factors based on a comprehensive search of original studies on the occurrence of social isolation and related influencing factors in PLWH. The aim is to reveal the prevalence of social isolation among PLWH and its related influencing factors, so as to provide data support for further clinical identification of the high-risk group of PLWH in whom social isolation occurs, and to provide a more comprehensive pre-study basis for its etiological study.

## Materials and methods

### Study registration

This system review will be carried out in accordance with the Preferred Reporting Project (PRISMA-P) statement guidelines (S1 File) for systematic reviews and meta-analysis

programs,and has been registered on the system review registration platform Prospero, registration number: CRD42024499044.

## Inclusion and exclusion criteria

The inclusion and exclusion criteria in this review were formulated based on the PECOS (Population, Exposure, Comparison, Outcome, and Study design) principles.

## PECOS description

**Population.** People living with HIV.

**Exposure.** Influential factors of social isolation. Influencing factors are exposures that increase the incidence of social isolation among PLWH, may include: age, gender, marital status, education, cognitive reserve, income, etc.

**Comparator.** Reference groups were established for each influencing factor in every study, such as the prevalence of social isolation among male and female PLWH, as well as the prevalence of social isolation among married, unmarried, and divorced PLWH.

**Outcome.** The prevalence of social isolation and its influence factors or determinants among PLWH.

**Study type.** Observational studies, including cross-sectional, case-control, and cohort designs, that reported on the incidence or risk factors for social isolation among PLWH Will be included.

**Inclusion criteria.**

I.  *Participant*. We will incorporate studies involving PLWH who have been diagnosed as social isolation according to any recognized diagnostic criteria, such as the Lubben Social Network Scale [22], regardless of gender, age, race, nationality, or occupation.

II. *Type of study*. Published prospective or retrospective studies on observational studies (including cross-sectional, cohort and case-control studies) which report on the incidence of social isolation among PLWH and the influence factors.

III. *Type of exposure*. Determinants of social isolation.

IV. *Types of outcomes*. Primary outcome indicators included different population characteristics (gender, age, occupation, education, etc.), the prevalence of social isolation among HIV patients in different regions, and risk factors associated with the occurrence of social isolation among HIV patients.

Secondary outcomes included general characteristics of the patients (region, age, sex, ethnicity), time of investigation, diagnostic criteria, and type of study.

**Exclusion criteria.**

I.  Studies with incomplete information and data that could not be extracted.

II. Studies reported or published repeatedly in the same study population, excluding lower quality studies.

III. Studies not in Chinese or English.

IV. Studies for which only the abstract was available and the full text could not be obtained by contacting the authors.

V.  Lower quality studies (AHRQ scores, Newcastle Ottawa scale scores below 3 points).

## Data sources and search strategy

The following 10 databases will be searched: English databases (PubMed, EMBASE, Web of Science, The Cochrane Library, CINAHL, Google Scholar), Chinese databases (China National Knowledge Infrastructure (CNKI), Wanfang Data, China Biomedical Literature Database (CBM), China Science and Technology Journal Database (VIP). The search was conducted from the database establishment to January 2024. In order to improve the sensitivity, we used a combination of subject terms and free text terms to retrieve the literature. Subject terms included: Social Isolation, HIV, Acquired Immunodeficiency Syndrome, Epidemiology, incidence, Risk Factors. The free word included: Social Exclusion, Social Separation, Social segregation, Social seclusion, Social shielding, Social partitioning, Social solitariness, Social aloneness, Human Immunodeficiency Virus, Lymphadenopathy Associated Virus, AIDS, HIV, Epidemiology, frequency, occurrence, outbreaks, prevalence, incidence, Risk Factor, influence, Affecting factor and so on. At the same times we will search the bibliographies of all studies included in the systematic review to ensure that no relevant studies are missed. The specific search strategy employed in this study is outlined in S2 File, taking Pubmed as an illustrative example.

## Study selection

Literature screening will follow the following steps:

I. Use EndNote X9 software to find the duplicates imported after searching, and delete the duplicates among databases and record them after comparing the information of authors, titles, journal names, publication time, keywords, abstracts, etc. one by one.

II. Two researchers independently and carefully read the titles and abstracts of all the retrieved literature to determine the relevance of the literature to the research questions of the systematic evaluation, exclude obviously irrelevant literature and indicate the reasons for exclusion, and proceed to full-text screening of literature that is not sure of its relevance.

III. To obtain the full text of the potentially qualified and uncertain literature initially selected, two researchers independently carefully read and evaluate the full text of the literature, extract the relevant information in the literature, determine whether the literature meets the inclusion criteria of systematic evaluation, and decide whether the literature is included.

Note: Throughout the screening of the literature, two researchers were required to independently screen the literature, make comparisons at the end of each step, and seek arbitration from a third researcher in case of disputes.

## Data extraction

Data extraction will be carried out according to a pre-designed data extraction form. Data and information management will be carried out using Microsoft Excel 2016.

Two researchers read the full text of the included literature in detail and independently extracted all relevant information from the study, including basic information of the included literature, basic characteristics of the study and risk factors. Check the extracted data, and if there is any disagreement, check the original literature and modify it.

I. Basic information of the included literature: including the title of the literature, the name of the first author, journal information (journal name, year, volume, issue and page number), research location, extraction time and extraction personnel.

II.  Basic research information: Study type, study time, study country, data collection method (clinical interview or questionnaire), sample size, number of people with social isolation, prevalence rate of social isolation, characteristics of study subjects (including age, gender, income, education level, sexual behavior characteristics, diagnostic criteria for social isolation, etc.), and disease characteristics (diagnosis method of HIV, course of disease, and treatment method) Type, etc.).

III.  The study of risk factors should also extract: The total number of patients in the case group and the control group, the number of positive social isolation, the number of negative social isolation, the mean and standard deviation of risk factors related to the case group and the control group respectively.

The above literature screening process will be carried out according to PRISMA-P [23], as shown in Fig 1.

## Risk of bias and quality assessment

The cross-sectional study will be assessed for literature quality using 11-item instrument recommended by the Agency for Healthcare Research and Quality [24].

Each item was answered with "yes," "no," or unclear. If the answer is "no" or "unclear", the item will receive a score of "0"; If the answer is "yes", the item is scored as "1". The quality of literature was evaluated as follows: low quality = 0~3; Medium quality = 4~7; High quality = 8~11.

Cohort studies and case-control studies were evaluated by the Newcastle-Ottawa scale recommended by the Non-randomized Research Methods Group of the Cochrane Collaboration [25] n. The scale consists of eight items in three aspects: group selection, comparability between groups, and measurement of exposure or outcome. The total score of each item was 1, except for the item of comparability between groups 2 points. The scores of literature quality ranged from 0 to 9. The literature with scores of $\leq$3, 4–6 and $\geq$7 were defined as low quality, medium quality and high quality, respectively.

Two researchers independently assessed literature quality and cross-checked their results. In cases where their conclusions differed, consensus was reached through discussion.

## Data analysis and assessment of publication bias

**Data analysis.**  The Stata 15.1 software will be used for data analysis.

Ethics and dissemination Heterogeneity between the included studies was judged by Q test ($P$ value) and $I^2$. If $I^2$ was <50% and $P$ of Q test was >0.1, it was considered that there was good homogeneity among the studies. When there was no obvious heterogeneity among the studies, the fixed effect model was used to calculate the pooled incidence of social isolation of HIV patients. Otherwise ($P\leq$0.1 or $I^2\geq$50%), it was considered that there was significant heterogeneity among the studies, and the random effects model was used for the combined calculation. If there was significant heterogeneity among the included studies, the following measures would be taken: the data were checked again; Subgroup analysis and other methods were used to explore the sources of heterogeneity; After excluding the studies with abnormal results, the meta-analysis was performed again, and the results were compared with the meta-analysis without abnormal results to explore the impact of the study on the combined effect; The heterogeneity among studies was too obvious, and the source of heterogeneity was explored by subgroup analysis and sensitivity analysis. The inverse variance method or Mantel-Haenszel method was used to calculate the OR or mean difference of the influencing factors of social isolation in HIV patients. If the data from the included studies could not be synthesized quantitatively, a statistical description was conducted.

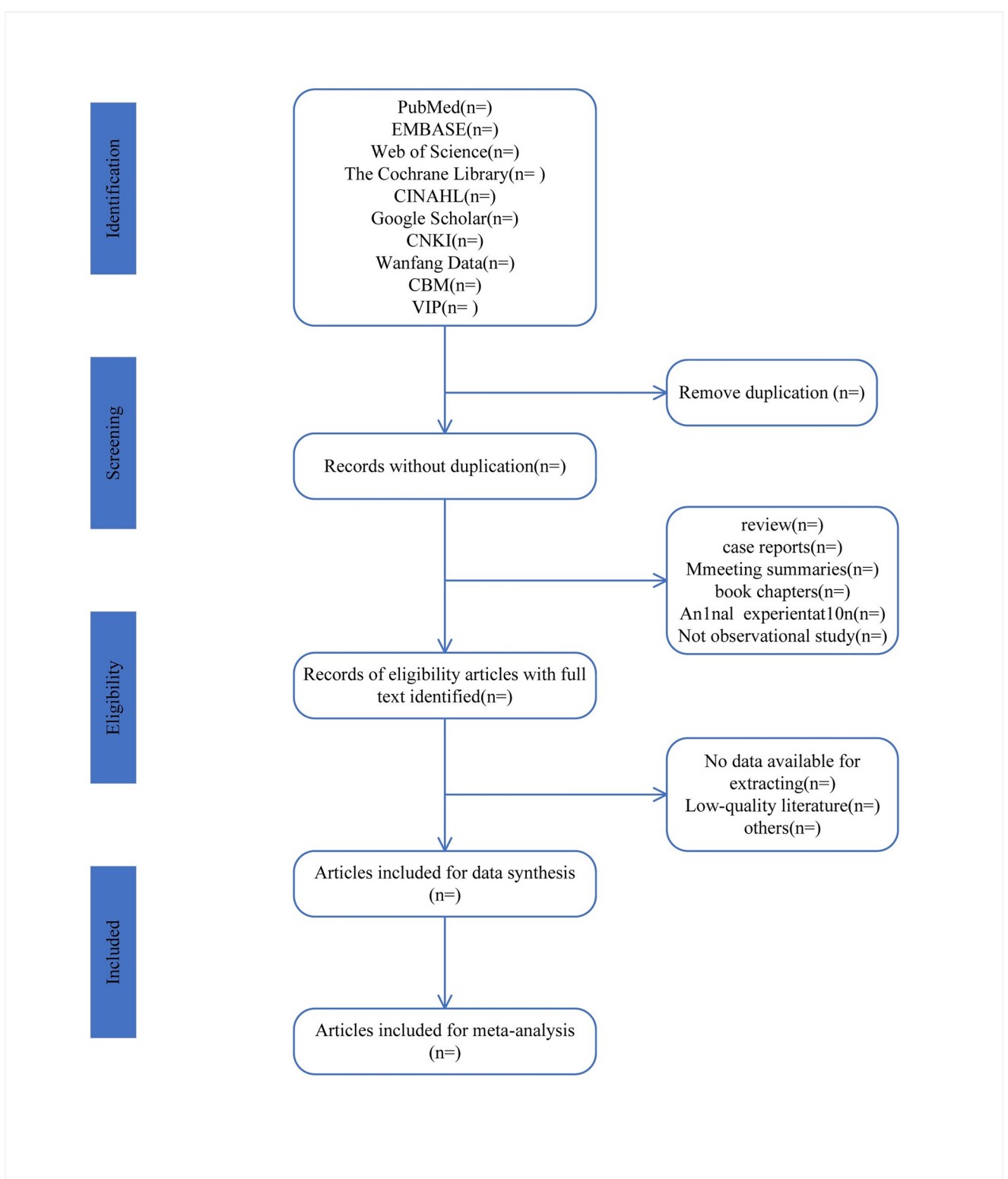

**Fig 1. Flow diagram of study selection process.** (n): is the number of articles that will be included at each stage.

**Publication bias.**   The funnel plot was visually assessed to detect potential publication bias. In the presence of noticeable asymmetry, it suggests the existence of certain publication bias. Begg's rank correlation test and Egger's linear regression method were employed for quantitative evaluation of publication bias in the meta-analysis investigating the incidence of social isolation and its associated influencing factors among PLWH.

Sensitivity analysis was carried out by excluding the studies with small sample size, large weight and high risk of bias one by one, then the meta-analysis was performed again, and the results were compared with the Meta-analysis without excluding abnormal results to explore the impact of the study on the combined effect.

## Ethics and dissemination

This study is an overview based on published articles, so there is no need for ethical review. The study will be conducted in accordance with the protocol approved by PROSPERO in January 2024: the database search will commence on January 25, 2024, and the entire review process is anticipated to conclude by June 30, 2025.

This system evaluation will be published in peer-reviewed journals and the review will be circulated in a peer-reviewed journal or conference report.

## Results of the study

The results of this study will shed light on the prevalence of social isolation among PLWH by comparing the differences in the incidence of social isolation among PLWH by different population characteristics (gender, age, education, marital status, etc.), by country/region, and by other relevant factors (duration of HIV disease, use of antiretroviral medication, etc.), the differences in the incidence of social isolation in female versus male HIV patients, thus helping to identify the groups at risk for social isolation. This will reveal the prevalence of social isolation among HIV patients, and thus help to identify groups at high risk of social isolation among HIV patients.

At the same time, the results of this study will also reveal the risk factors related to HIV social isolation through Meta-analysis, so as to provide a basis for clinical targeted scientific prevention programs and provide an important reference for the formulation of public health policies.

## Discussion

Social isolation is associated with a variety of adverse health outcomes: it not only increases the risk of falls, cognitive decline, coronary heart disease, depression, stroke, etc., but also increases the all-cause mortality of patients [26–30]. However, existing evidence indicates that PLWH are more prone to experiencing social isolation as a result of stigma and discrimination, which can have numerous detrimental effects on their well-being.

Understanding the prevalence of social isolation among PLWH and related influencing factors is an important prerequisite for effective intervention.

However, because of the limitation of sample size, funds and the regional difference is the lack of a representative study of social segregation occurred among people infected with HIV. In addition, no meta-analysis has been conducted on the current status of social isolation and related influencing factors in this population. Therefore, this study compares the social isolation of PLWH patients with different characteristics (such as age, gender, marital status, education level, time of HIV diagnosis, etc.) by meta-analysis, which can not only comprehensively understand the prevalence of social isolation in PLWH, but also explore the related influencing factors of social isolation in PLWH. Important indicators to identify high-risk groups.

Limitations of the study. Firstly, due to the limitation of language factors, only Chinese and English literature was included in this paper, while literature published in other languages was not included in the analysis, which may lead to some bias in the results. Secondly, for the meta-analysis part of the incidence of social isolation in PLWH, there may be a large heterogeneity among the literature due to the limitation of the characteristics of the individual rate Meta-analysis (in order to explore the source of the heterogeneity, subgroup analyses based on factors such as gender, age, and so on will be conducted in this study).

## Supporting information

**S1 File. PRISMA-P 2015 checklist.**
(DOCX)

**S2 File. Search strategy of Pubmed database.**
(DOCX)

## Author Contributions

**Conceptualization:** Qiao Wu.

**Data curation:** Qiao Wu, Shu Chen.

**Formal analysis:** Qiao Wu, Shu Chen.

**Investigation:** Shu Chen, Lingling Jiang.

**Project administration:** Qiao Wu.

**Software:** Qiao Wu, Jiayi Wang, Xiaogang Liao.

**Supervision:** Jiarong Tan.

**Validation:** Jiayi Wang, Xiaogang Liao, Lingling Jiang.

**Visualization:** Jiayi Wang.

**Writing – original draft:** Qiao Wu, Shu Chen.

**Writing – review & editing:** Qiao Wu.

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
