## [Decision Letter · Decision Letter 0]

10 Apr 2024

PONE-D-24-03090Incidence and influencing factors related to social isolation among HIV/AIDS patients: protocol for a systematic review and meta-analysisPLOS ONE

Dear Dr. Tan,

Thank you for submitting your manuscript to PLOS ONE. After careful consideration, we feel that it has merit but does not fully meet PLOS ONE’s publication criteria as it currently stands. Therefore, we invite you to submit a revised version of the manuscript that addresses the points raised during the review process. **This study is a useful addition to the current literature regarding social isolation among people living with HIV. The overall design is appropriate. There are several aspects of the paper that need improvement. Specifically, the description of approach and methods needs refinement. For example, the period of the study is not stated, inclusion and exclusion criteria are unclear, and data sources and search strategy need further details. There is also some confusion about the place of results, discussion, and conclusions in the paper. Some reorganization and streamlining there is needed. Please consider and respond to all comments provided by both reviewers, and revise the paper accordingly.** Please submit your revised manuscript by May 25 2024 11:59PM. If you will need more time than this to complete your revisions, please reply to this message or contact the journal office at plosone@plos.org. Please include the following items when submitting your revised manuscript:A rebuttal letter that responds to each point raised by the academic editor and reviewer(s). You should upload this letter as a separate file labeled 'Response to Reviewers'.A marked-up copy of your manuscript that highlights changes made to the original version. You should upload this as a separate file labeled 'Revised Manuscript with Track Changes'.An unmarked version of your revised paper without tracked changes. You should upload this as a separate file labeled 'Manuscript'.

We look forward to receiving your revised manuscript.

Kind regards,

Magdalena Szaflarski, PhD

Academic Editor

PLOS ONE

Reviewers' comments:

Reviewer's Responses to Questions

**Comments to the Author**

1. Does the manuscript provide a valid rationale for the proposed study, with clearly identified and justified research questions?

Reviewer #1: Yes

Reviewer #2: Yes

2. Is the protocol technically sound and planned in a manner that will lead to a meaningful outcome and allow testing the stated hypotheses?

Reviewer #1: Yes

Reviewer #2: Yes

3. Is the methodology feasible and described in sufficient detail to allow the work to be replicable?

Reviewer #1: Yes

Reviewer #2: Yes

4. Have the authors described where all data underlying the findings will be made available when the study is complete?

Reviewer #1: Yes

Reviewer #2: Yes

5. Is the manuscript presented in an intelligible fashion and written in standard English?

Reviewer #1: Yes

Reviewer #2: Yes

6. Review Comments to the Author

You may also provide optional suggestions and comments to authors that they might find helpful in planning their study.

**Reviewer #1**: Dear authors,

First of all thanks for your manuscript called Incidence and influencing factors related to social isolation among HIV/AIDS patients: protocol for a systematic review and meta-analysis. I found really valuable the contribution of this paper in the field, it is systematic review and meta-analysis to systematically identify and synthesize the existing literature on the occurrence of social isolation in HIV patients and its associated influencing factors. Anyways I have some concerns with the content and hope the feedback can be taken into consideration and will help with the manuscript quality.

Abstract

1.Introduction should not appear in the abstract, and I can suggest to the authors that they draw the research topic in the background. (Line 11-17)

2.Authors should adjust the structure and write background, methods, results and conclusions in the abstract. (Line 11-36)

3.The sentence “This will be the first systematic review......and its associated influencing factors” is the research significance, it does not belong to the research conclusion. (Line 30-32)

Introduction

Introduction need a restructuring to make all the idea clear, I suggest to use the second-level title to separate theoretical concepts from other contents, such as social isolation among PLWH.

Materials and methods

1.The introduction part has indicated the purpose of the study (Line 68-72), and the research method need not repeat the objectives. (Line 74-79) I suggest to keep the former, because data acquisition, data coding, data analysis should be main content in materials and methods section.

2.The authors say that social isolation is a common phenomenon among adults in introduction, for example, “Social isolation is prevalent psychosocial processes among adults.” (Line 12) “Studies have demonstrated that more than one in ten 48 individuals in the adult population reported social isolation and nearly one-quarter 49 of adults.” (Line 48-51) But according to the screening criteria, the population is included in any age. “We will include studies of PLWH who have been diagnosed as social isolation according to any recognized diagnostic criteria, regardless of age.” (Line 88) I suggest that authors add social isolation phenomenon of other age groups in the introduction section for smooth cohesion.

Inclusion and exclusion criteria

1.Author should use as objective a description as possible, “any recognized diagnostic criteria” is too ambiguous, you may give specific examples, like based on some widely used scale. (Line 88)

2. I recommend to add a PICO process just to make the search process more clear. Population, Intervention, Comparison(s) and Outcome (PICO) is usually used for systematic review and meta-analysis of clinical trial study. For the study without Intervention or Comparison(s), it is enough to use P (Population) and O (Outcome) only to formulate a research question. A well-formulated question creates the structure and delineates the approach to defining research objectives. (Line 87-98)

Data sources and Search strategy

1.The selected databases represent the quality of the articles included in the meta-analysis, the authors mainly search several databases containing high quality papers (like Web of Science). In order to retrieve the suitable and substantial quantitative empirical research studies, a systematic and comprehensive search was conducted, we use Google Scholar and so on as a supplementary database. And what is the total number of articles retrieved that met the criteria? (Line 107-110)

2.Literature search should clearly show the screening process of English database and Chinese database with PRISMA flow chart, and the number of articles included in the meta-analysis should be presented according to the screening criteria. (Line 122)

Data extraction

1.The article should focus on the research process and results, and the researcher can divide the division in Contributors. (Line 144)

2.The article is missing a document coding table, which can clearly present the characteristics of the selected literature. (Line 157)

Risk of bias and quality assessment

The article should focus on the research process and results, and the researcher can divide the division in Contributors. (Line 177)

Data analysis and assessment of publication bias

The authors do not present the funnel plot, how the data points are distributed on the funnel plot? What does the distribution show on the funnel plot? (Line 201)

Discussion

The Results section is missing before Discussion, and the Conclusion chapter is missing after Discussion. The Result is an objective presentation of research result. In discussion, authors can write the association and difference between your research results and other studies to further highlight the significance of research, the limitations of research, and suggestions for subsequent studies and so on. Conclusion is a summary of the full text, authors can make a selective summary of the important content. (Line 206)

Minor edits

First letter of “Which” should use lower-case letter. (Line 40)

**Reviewer #2**: The manuscript appears formulaic. There is no study period. The authors also do not designate the types of studies they will focus on. How will they operationalize social isolation for this study. In addition, to incidence and prevalence they could also look at the impact of social isolation on various outcomes.

7. PLOS authors have the option to publish the peer review history of their article (what does this mean?). If published, this will include your full peer review and any attached files.

Reviewer #1: No

Reviewer #2: No

---

## [Author Response · Author response to Decision Letter 0]

2 Jul 2024

Dear Editors and Reviewers:

We sincerely appreciate your letter and the reviewers' insightful comments on our manuscript titled 'Incidence and influencing factors related to social isolation among HIV/AIDS patients: protocol for a systematic review and meta-analysis.' Your comments and those of the reviewers were highly insightful and enabled us to greatly improve the quality of our manuscript.In the following pages are our point-by-point responses to each of the comments of the reviewers as well as your own comments.

Revisions in the text are shown using yellow highlight for additions, and strikethrough font [example] for deletions. 

Below is the opinion of the referees, we make a point by point response to, and points out the revision.

Responds to the reviewer’s comments:

Reviewer #1:

1.Abstract

1.1Comment:Introduction should not appear in the abstract, and I can suggest to the authors that they draw the research topic in the background. (Line 11-17)

1.1Reply:We express our gratitude to the reviewer for bringing this matter to our attention.In accordance with your suggestion,we have adjusted the introduction in the abstract to the research backgroundand enhancing the corresponding content; please refer to lines 11-18 of the revised manuscript.

1.2Comment:Authors should adjust the structure and write background, methods, results and conclusions in the abstract. (Line 11-36)

1.2Reply:We express our gratitude to the reviewer for bringing this matter to our attention. In response, we have restructured the abstract into four sections: background, methods and analysis, results, and conclusions. Furthermore, we have made corresponding revisions in content as well. Please refer to lines 11-39 of the revised manuscript for further details.

1.3Comment:The sentence “This will be the first systematic review......and its associated influencing factors” is the research significance, it does not belong to the research conclusion. (Line 30-32)

1.3Reply:We express our gratitude to the reviewer for bringing this matter to our attention. In response, we have made revisions to the abstract section of the study; kindly refer to lines 34-39 in the revised manuscript.

2.Introduction

2.1Comment:Introduction need a restructuring to make all the idea clear, I suggest to use the second-level title to separate theoretical concepts from other contents, such as social isolation among PLWH.

2.1Reply:We appreciate the reviewer for bringing this to our attention. In response to your suggestion, we have revised the introduction section accordingly.The revised introduction will encompass the following aspects: the definition of social isolation, the prevalence of social isolation, the fact that HIV patients are more likely to be socially isolated due to a variety of reasons (e.g., discrimination and stigma, etc.), the prevalence of social isolation among HIV patients (which can be as high as 48.5% to 59%), and the harms of social isolation to HIV patients, The current status of research on social isolation among HIV patients (no large-scale epidemiologic studies, no meta) and the significance of this study. See lines 43-80 of the revised manuscript for details.

3.Materials and methods

3.1Comment:The introduction part has indicated the purpose of the study (Line 68-72), and the research method need not repeat the objectives. (Line 74-79) I suggest to keep the former, because data acquisition, data coding, data analysis should be main content in materials and methods section.

3.1Reply:We appreciate the reviewer for bringing this to our attention.The purpose of the study has been omitted from the Materials and Methods section in accordance with your suggestion.

3.2Comment:The authors say that social isolation is a common phenomenon among adults in introduction, for example, “Social isolation is prevalent psychosocial processes among adults.” (Line 12) “Studies have demonstrated that more than one in ten 48 individuals in the adult population reported social isolation and nearly one-quarter 49 of adults.” (Line 48-51) But according to the screening criteria, the population is included in any age. “We will include studies of PLWH who have been diagnosed as social isolation according to any recognized diagnostic criteria, regardless of age.” (Line 88) I suggest that authors add social isolation phenomenon of other age groups in the introduction section for smooth cohesion.

3.2Reply:We appreciate the reviewer for bringing this to our attention.According to your suggestion, the following modifications have been made in the introduction: the previous content "social isolation is a common phenomenon in adults" has been revised to "Nowadays, the incidence of social isolation is increasing"; Additional description on the incidence of juvenile social isolation.See lines 41-53 of the revised manuscript for details.

4.Inclusion and exclusion criteria

4.1Comment:Author should use as objective a description as possible, “any recognized diagnostic criteria” is too ambiguous, you may give specific examples, like based on some widely used scale. (Line 88)

4.1Reply：We appreciate the reviewer for bringing this to our attention. According to your suggestion, we have made amendments to the diagnostic criteria for Social isolation and included the widely recognized Lubben Social Network Scale as an illustrative example. For further information, please refer to the revised section 106-107.

4.2Comment:I recommend to add a PICO process just to make the search process more clear. Population, Intervention, Comparison(s) and Outcome (PICO) is usually used for systematic review and meta-analysis of clinical trial study. For the study without Intervention or Comparison(s), it is enough to use P (Population) and O (Outcome) only to formulate a research question. A well-formulated question creates the structure and delineates the approach to defining research objectives. (Line 87-98)

4.2Reply:We appreciate the reviewer for bringing this to our attention. According to your suggestion, we have incorporated the PECOS process into the inclusion and exclusion criteria section. For further details, please refer to the revised line 88-103.

5.Data sources and Search strategy

5.1Comment:The selected databases represent the quality of the articles included in the meta-analysis, the authors mainly search several databases containing high quality papers (like Web of Science). In order to retrieve the suitable and substantial quantitative empirical research studies, a systematic and comprehensive search was conducted, we use Google Scholar and so on as a supplementary database. And what is the total number of articles retrieved that met the criteria? (Line 107-110)

5.1Reply:This study is a protocol of systematic review and meta-analysis. Therefore, there is no comprehensive literature screening at this stage. The current retrieval results show that 10 Chinese and English databases were searched, and a total of 3133 literatures were retrieved, including: CNKI(n=215); WanFang Data(n=175); VIP(n=76); CBM (n=284); Pubmed(n=372); Web of Science(n=461); Embase(n=108); CENTRAL(n=63); The Cochrane Library (n=82); Google Scholar (n=1297).

5.2Comment:Literature search should clearly show the screening process of English database and Chinese database with PRISMA flow chart, and the number of articles included in the meta-analysis should be presented according to the screening criteria. (Line 122)

5.2Reply:We appreciate the reviewer for bringing this to our attention. According to your suggestion, we drew a flow chart of the literature screening process according to PRISMA-P, as shown in Figure 1.

6.Data extraction

6.1Comment:The article should focus on the research process and results, and the researcher can divide the division in Contributors. (Line 144)

6.1Reply:We appreciate the reviewer for bringing this to our attention. According to your suggestion, the role of the investigator in data extraction was removed and only stated in the investigator contributions.

6.2Comment:The article is missing a document coding table, which can clearly present the characteristics of the selected literature. (Line 157)

6.2Reply:We appreciate the reviewer for bringing this to our attention. Because this study is a protocol of systematic review and meta-analysis . At present, there is no specific coding table for the included literature to present the characteristics of the selected literature. The results of the subsequent included literature will be presented in the complete meta-analysis to be published later.

7.Risk of bias and quality assessment

7.1Comment:The article should focus on the research process and results, and the researcher can divide the division in Contributors. (Line 177)

7.1Reply:We appreciate the reviewer for bringing this to our attention. Same with the process of information extraction, according to your advice, we will be the researchers in the risk of bias and the role of quality assessment for deleted, only make notes in the contribution.

8.Data analysis and assessment of publication bias

8.1Comment:The authors do not present the funnel plot, how the data points are distributed on the funnel plot? What does the distribution show on the funnel plot? (Line 201)

8.1Reply:We appreciate the reviewer for bringing this to our attention. This study provides additional instructions on how to determine the existence of publication bias based on the funnel plot. For details, please refer to lines 226-230 of the revised manuscript. However, this study is a systematic review and meta-analysis scheme, and the funnel plot cannot be presented at this stage. The specific publication bias results of this study will be presented in the complete meta-analysis to be published later.

9.Discussion

9.1Comment:The Results section is missing before Discussion, and the Conclusion chapter is missing after Discussion. The Result is an objective presentation of research result. In discussion, authors can write the association and difference between your research results and other studies to further highlight the significance of research, the limitations of research, and suggestions for subsequent studies and so on. Conclusion is a summary of the full text, authors can make a selective summary of the important content. (Line 206)

9.1Reply:We appreciate the reviewer for bringing this to our attention. According to your advice, we increased the results part by the end of the discussion, and to discuss the corresponding modification, such as increasing research possible limitations and further highlights the significance of research. See lines 235-273 of the revised version for details.

Reviewer #2:

1.Comment: The manuscript appears formulaic. 

1.Reply:We appreciate the reviewer for bringing this to our attention. Due to the limitation of language level, this study language may be relatively smooth, to a certain extent, we also pay attention to this problem, the follow-up if need we will add the English speakers in the study the researchers, and the language of the article further polished.

2.Comment: There is no study period.

2.Reply:We appreciate the reviewer for bringing this to our attention.According to your suggestion, we have added the study period of the article, please refer to lines 276-278 of the revised manuscript for details.

3.Comment: The authors also do not designate the types of studies they will focus on. How will they operationalize social isolation for this study. 

3.Reply:This study focuses on observational studies (including cross-sectional, cohort and case-control studies). The inclusion criteria have been explained in the literature inclusion criteria, and please refer to lines 109-111 of the revised manuscript for details. And according to the recommendations for further clarifications research retrieval process, we added the PECOS process, the study of cover type further illustrates, details please refer to the revised line 103-101.

4.Comment: In addition, to incidence and prevalence they could also look at the impact of social isolation on various outcomes.

4.Reply:According to your suggestion, we have made additional comments, such as comparing the difference in the incidence of social isolation between female and male HIV patients, which can not only reveal the prevalence of social isolation in HIV patients, but also help to identify high-risk groups of social isolation in HIV patients. For details, please refer to lines 236-247 of the revised version.

Responds to the Academic Editor

1.Comment: The period of the study is not stated.

1.Reply: We have added the study period of the article, please refer to lines 276-278 of the revised manuscript for details.

2.Comment: Inclusion and exclusion criteria are unclear.

2.Reply: As recommended, the PECOS process was added to further clarify the inclusion and exclusion criteria, please refer to the revised line 88-103.

3.Comment: Data sources and search strategy need further details. 

3.Reply: A flow chart of the literature selection process was drawn according to PRISMA-P to further clarify the details of literature selection.

4.Comment: Here is also some confusion about the place of results, discussion, and conclusions in the paper. Some reorganization and streamlining there is needed. Please consider and respond to all comments provided by both reviewers, and revise the paper accordingly.

4.Reply:According to the suggestions, we reconstructed the results, discussions, and conclusions of the study, please refer to the revised line 235-273.

We would like to thank the referee again for taking the time to review our manuscript.

Yours sincerely

---

## [Editor Report · Decision Letter 1]

10 Jul 2024

Incidence and influencing factors related to social isolation among HIV/AIDS patients: protocol for a systematic review and meta-analysis

PONE-D-24-03090R1

Dear Dr. Tan,

We’re pleased to inform you that your manuscript has been judged scientifically suitable for publication and will be formally accepted for publication once it meets all outstanding technical requirements.

Kind regards,

Magdalena Szaflarski, PhD

Academic Editor

PLOS ONE

Additional Editor Comments (optional):

Thank you for your careful responses to the prior critiques and manuscript revision. No additional changes are recommended.
---

## [Editor Report · Acceptance letter]

15 Jul 2024

PONE-D-24-03090R1 

PLOS ONE

Dear Dr. Tan, 

I'm pleased to inform you that your manuscript has been deemed suitable for publication in PLOS ONE. Congratulations! Your manuscript is now being handed over to our production team.

Kind regards, 

on behalf of

Dr. Magdalena Szaflarski 

Academic Editor

PLOS ONE